# Vitamin D Metabolites in Nonmetastatic High-Risk Prostate Cancer Patients with and without Zoledronic Acid Treatment after Prostatectomy

**DOI:** 10.3390/cancers14061560

**Published:** 2022-03-18

**Authors:** Carsten Stephan, Bernhard Ralla, Florian Bonn, Max Diesner, Michael Lein, Klaus Jung

**Affiliations:** 1Department of Urology, Charité-Universitätsmedizin Berlin, 10115 Berlin, Germany; bernhard.ralla@charite.de; 2Berlin Institute for Urologic Research, 10115 Berlin, Germany; 3Immundiagnostik AG, 64625 Bensheim, Germany; florian.bonn@immundiagnostik.com (F.B.); max.diesner@immundiagnostik.com (M.D.); 4Department of Urology, Sana Medical Center Offenbach, 63069 Offenbach am Main, Germany; michael.lein@sana.de

**Keywords:** prostate cancer, prostatectomy, zoledronic acid treatment, vitamin D, circulating vitamin D metabolites, 25(OH)D_3_, 24,25(OH)_2_D_3_, 1,25(OH)_2_D_3_

## Abstract

**Simple Summary:**

Recent research on prostate cancer and vitamin D is controversial. We measured three vitamin D_3_ metabolites in 32 selected prostate cancer patients after surgery at four time points over four years. Within a large European study, half of the patients were prophylactically treated with zoledronic acid (ZA); the others received a placebo. After the study start, all the patients daily took calcium and vitamin D_3_. The development of metastasis was not affected by ZA treatment. While two vitamin D metabolites had higher values after the study’s start, with constant follow-up values, the 1,25(OH)_2_-vitamin D_3_ concentrations remained unchanged. The latter form was the only metabolite that was higher in the patients with metastasis as compared to those without bone metastasis. This result is surprising. However, it is too premature to discuss possible prognostic value yet. Our results should be confirmed in larger cohorts.

**Abstract:**

There are limited and discrepant data on prostate cancer (PCa) and vitamin D. We investigated changes in three vitamin D_3_ metabolites in PCa patients after prostatectomy with zoledronic acid (ZA) treatment regarding their metastasis statuses over four years. In 32 patients from the ZEUS trial, 25(OH)D_3_, 24,25(OH)_2_D_3_, and 1,25(OH)_2_D_3_ were measured with liquid chromatography coupled with tandem mass spectrometry at four time points. All the patients received daily calcium and vitamin D_3_. Bone metastases were detected in 7 of the 17 ZA-treated patients and in 5 of the 15 controls (without ZA), without differences between the groups (*p* = 0.725). While 25(OH)D_3_ and 24,25(OH)_2_D_3_ increased significantly after the study’s start, with following constant values, the 1,25(OH)_2_D_3_ concentrations remained unchanged. ZA treatment did not change the levels of the three metabolites. 25(OH)D_3_ and 24,25(OH)_2_D_3_ were not associated with the development of bone metastases. In contrast, 1,25(OH)_2_D_3_ was also higher in patients with bone metastasis before the study’s start. Thus, in high-risk PCa patients after prostatectomy, 25(OH)D_3_, 24,25(OH)_2_D_3_, and 1,25(OH)_2_D_3_ were not affected by supportive ZA treatment or by the development of metastasis over four years, with the exception of 1,25(OH)_2_D_3_, which was constantly higher in metastatic patients. There might be potential prognostic value if the results can be confirmed.

## 1. Introduction

Several international conferences in recent years have discussed, in detail, the current evidence and the ongoing controversies in vitamin D research [1,2,3]. Vitamin D_3_ is mainly formed in the skin from 7-dehydrocholesterol upon ultraviolet B exposure. It is subsequently hydroxylated by two cytochrome-P450-mediated hydroxylation processes. In the liver, it is converted to 25-hydroxyvitamin D_3_ (25(OH)D_3_), released in the bloodstream and subsequently hydroxylated in the kidney, but also in other organs, including the prostate, to 1,25-dihydroxyvitamin D_3_ (1,25(OH)_2_D_3_). The controversies concern both 25(OH)D_3_ as the primary circulating vitamin D form reflecting the vitamin D status and 1,25(OH)_2_D_3_, the actual active metabolite that reacts with the vitamin D receptor. The controversial issues relate particularly to the extraskeletal actions of vitamin D and involve numerous diseases, such as various malignancies and cardiovascular, dermatologic, and immunological disorders [4]. Cell research and animal studies provide strong evidence of the potential molecular and cellular mechanisms underlying the actions of vitamin D and its metabolites [5,6,7]. However, corresponding observational and randomized controlled studies, probably due to their weak study designs, have frequently remained inconclusive and showed conflicting results concerning the hypothesized beneficial effect of vitamin D [3,8]. In this respect, prostate cancer (PCa) is no exception.

The complex biochemical and molecular relationship between PCa and vitamin D has been reviewed in numerous reports [9,10,11]. The limited and also partly discrepant data regarding the action of vitamin D in the treatment of PCa are exemplarily reflected by the results reported in some pertinent following studies. For example, a meta-analysis from 21 studies published until 2013 revealed a significant 17% higher risk of PCa in men with higher serum levels of 25(OH)D_3_ [12]. In contrast, other studies reported that higher 25(OH)D_3_ levels were associated with a 57% reduction in the risk of lethal PCa or improved prognosis [13,14,15]. A recently performed dose–response meta-analysis of 25(OH)D_3_, which was based on seven relevant studies, supported the idea that a higher serum 25(OH)D_3_ concentration was an important protective factor in PCa progression and was associated with reduced PCa mortality [16]. However, other studies showed that 25(OH)D_3_ concentrations and vitamin D supplementation were not significantly associated with an increased PCa incidence and mortality rate [17,18,19]. An inverse association between the post-treatment plasma 1,25(OH)_2_D_3_ levels and all-cause and PCa-specific mortality in men with aggressive PCa suggested a possible beneficial effect of vitamin D supplementation in these men [20]. A recent study in 2021 found that men with PCa and vitamin D deficiency had higher overall and PCa-specific mortality, but there was no association between the risk of PCa (in biopsied men) and different vitamin D categories [21]. Our own data for 480 biopsied men also showed no correlation between 25(OH)D_3_ and the pathological Gleason grade or differences between 222 men with and 258 without PCa [22].

It can be assumed that these contradictory results are mainly because basic principles for studies on the effect of vitamin D concerning health status have frequently been disregarded. Amrein et al. [8] defined four preconditions for an optimal study design in their seminal article “Vitamin D deficiency 2.0” as follows: (a) the measurement of the vitamin D status at baseline, (b) the consideration of vitamin D deficiency as a study inclusion criterion, (c) the application of an intervention capable of altering the vitamin D status, and (d) repeat measurements to verify the vitamin D status.

Considering these indispensable aspects for a valid study, it was therefore of interest to find out the possible changes in vitamin D status in patients after prostatectomy under the influence of the bisphosphonate zoledronic acid (ZA). ZA was shown to prevent bone loss due to antiandrogen-deprivation therapy, reduce morbidity and pain, and improve survival in castration-resistant PCa [23,24]. ZA induces the direct inhibition of PCa cells in vitro, inhibits tumor-mediated angiogenesis, enhances bone-mineral density, and suppresses bone markers [25,26,27]. Recent guidelines regarding PCa management recommend ZA and Denosumab as bone-protective agents in the supportive care of patients with castration-resistant PCa and skeletal metastases to prevent or reduce skeletal-related events [28,29,30,31]. However, data on the vitamin D status in follow-up measurements from patients receiving ZA are rare. They mostly refer to patients suffering from osteoporosis, while detailed data from PCa patients are lacking [32,33].

The basis for the study on vitamin D metabolites presented here was the availability of serum samples from a randomized, open-label study to evaluate the efficacy of ZA treatment for bone-metastasis prevention in high-risk PCa patients [34]. Thus, we were able to initiate this study in a small subset of 32 patients to largely meet the above-described requirements for a valid vitamin D study measuring the three metabolites 25(OH)D_3_, 24,25(OH)_2_D_3_, and 1,25(OH)_2_D_3_. With this study, we intended to obtain better insights into the following open issues: (a) the changes in vitamin D metabolites in PCa patients after prostatectomy over four years, (b) possible ZA-treatment effects on the profile of vitamin D metabolites, and (c) abnormalities in the metabolite profile with regard to metastasis during the study.

## 2. Materials and Methods

### 2.1. Patients and Samples

The study was based on vitamin D measurements performed on blood samples available from PCa patients after radical prostatectomy in the ZEUS trial (https://www.isrctn.com/ISRCTN66626762 accessed on 6 February 2022; https://doi.org/10.1186/ISRCTN66626762). This trial was a randomized, open-label study to evaluate the efficacy of ZA treatment for bone-metastasis prevention in high-risk PCa patients [34]. Ethical approval was obtained from local medical ethics committees for all the participating hospitals of this multicenter study, and the patients signed an informed consent form. The details and results of this trial were previously reported [34]. Briefly, the here-investigated subgroup consisted of nonmetastatic PCa patients with at least one of three high-risk factors: a Gleason score of 8–10, node-positive disease, or prostate-specific antigen (PSA) at diagnosis ≥20 ng/mL. No other prior PCa treatment (antiandrogen monotherapy, chemotherapy, and treatment with bisphosphonates) was allowed. All the patients were included in this study within 6 months after radical prostatectomy. The patients either received an intravenous infusion of 4 mg every three months or were without ZA treatment and served as controls. All the patients were prescribed concomitant therapy with a daily 500 mg dose of calcium and 400–500 IU of vitamin D_3_. Blood samples were collected under standard conditions in BD Vacutainer tubes before the study began and at every three-month visit. Serum samples were prepared and frozen at −80 °C until analysis. We analyzed samples from 32 patients at four time points, as further explained in the Results.

### 2.2. Analytics for Vitamin D Metabolites

The 25(OH)D_3_ and 24,25(OH)_2_D_3_ concentrations were determined with the KM1320 assay, and the concentration of 1,25(OH)_2_D_3_ was determined with a development version of the KM1400 assay, both from Immundiagnostik AG, Bensheim, Germany. The vitamin D metabolites were purified by immunoaffinity enrichment, with 1,25(OH)_2_D_3_ additionally derivatized for improved detection, and subsequently analyzed by liquid chromatography–tandem mass spectrometry on a QTrap 5500 system coupled to an Exion LC (AB Sciex, Darmstadt, Germany). All the samples were analyzed in two replicates using individual 3-point linear calibration curves. All the calibrants and controls were prepared from certified reference material (Cerilliant Corp., Round Rock, TX, USA) and validated with NIST^®^SRM^®^972a samples, if available. For 1,25(OH)_2_D_3_, reference samples are not available, but the calibrants were tested with samples from the Vitamin D External Quality Assessment Scheme (DEQAS). The reproducibility of the measurements was calculated as the within-run precision from the duplicate measurements using the root-mean-square method [35]. The coefficients of variation (and their 95% confidence intervals) were 3.28% (2.92 to 3.76%) for 25(OH)D_3_, 4.73% (3.30 to 5.82%) for 24,25(OH)_2_D_3_, and 8.96% (6.90 to 10.6%) for 1,25(OH)_2_D_3_.

### 2.3. Statistical Analysis

MedCalc 20.027 (MedCalc Software, Ostend, Belgium) and GraphPad Prism 9.3.1 (GraphPad Software, La Jolla, CA, USA) were used as statistical programs as previously described [36]. One-way and two-way analyses of variance (ANOVAs) were performed. Repeated-measures analyses of variances (ANOVAs) were used for a single-factor study without a grouping variable or for a two-factor study with a specified grouping variable. Holm and Sidak’s multiple-comparison test was applied to account for multiple testing. Pearson correlation analysis was used to determine the strength of the associations between vitamin D_3_ metabolites. Two-sided *p*-values < 0.05 were considered statistically significant. The values in the figures are presented as the means ± 95% confidence intervals (95% CIs).

## 3. Results

### 3.1. Patient Characteristics and Study Design

The study included a total of 32 patients after radical prostatectomy characterized by at least one of three high-risk factors: a Gleason score of 8–10, node-positive disease, and PSA of ≥20 ng/mL at diagnosis. The individual data of all the patients are summarized in Appendix A. Nineteen patients exhibited one high-risk factor (2 × positive nodes, 5 × PSA, and 12 × Gleason score), twelve patients had two factors (2 × PSA plus Gleason, 3 × PSA plus positive nodes, and 7 × Gleason plus positive nodes), and one patient had all three factors. The study started for 16 patients each in winter/spring and summer/autumn (Figure 1). In every patient, repeated measurements of vitamin D metabolites were performed in serum samples taken at four time points: before the study entry as baseline, after 3 and 9 months, and between 27 and 47 months when the study ended or bone metastasis was diagnosed. ZA was administered to 17 patients; 15 patients were controls and did not receive ZA treatment. Bone metastases were detected in 7 of the 17 ZA-treated patients during the study and in 5 of the 15 controls, indicating no significant differences between the two patient groups (Fisher’s exact test, *p* = 0.725). This result corresponded with that of the ZEUS trial [34].

The three vitamin metabolites 25(OH)D_3_, 24,25(OH)_2_D_3_, and 1,25(OH)_2_D_3_ were analyzed in detail. 25(OH)D_2_ and 1,25(OH)_2_D_2_ were also measured, but in all 128 samples, the 25(OH)D_2_ concentrations were found to be under the lower limit of quantitation of 3.6 nmol/L, and 1,25(OH)_2_D_2_ was not detectable. Thus, only the results for the three vitamin D_3_ metabolites are reported here. The effects of the two abovementioned potential influencing factors “ZA treatment (yes/no)” and “bone metastasis during the study (yes/no)” as well as the seasonal dependency of the vitamin D_3_ status were evaluated.

### 3.2. Vitamin D_3_ Metabolites in the Total Study Cohort and Dependency on the Season of the Start of the Study

Figure 2a,c,e provide an overview of the concentration changes for the three metabolites in the total study cohort of the 32 patients during the study at four measuring points. Statistically significantly increased levels of 25(OH)D_3_ and 24,25(OH)_2_D_3_ were observed within three months after the study’s start, with approximately constant values at the two subsequent measuring points. In contrast, the 1,25(OH)_2_D_3_ concentrations remained statistically unchanged over the entire study period.

As the seasonal dependency of the vitamin D_3_ status, with lower concentrations in winter and spring in comparison to summer and autumn, is also well known for PCa patients [22,37,38], we subdivided the patients into two groups with respect to the season of their study entry (Figure 2b,d,f). Lower levels of 25(OH)D_3_ and 24,25(OH)_2_D_3_ were detected in the patients who started their study in winter/spring in comparison to the patients with a study start in summer/autumn. While the patients with a study start in winter/spring showed distinctly increased concentrations of the two metabolites after three months of treatment, only moderately increased levels were found in the patients with summer/autumn study entry (Figure 2b,d). For 1,25(OH)_2_D_3_, a subdivision of the patients did not have any effect on the influence of its concentration behavior over the entire study period (Figure 2f).

When evaluating the data, it must be taken into account that all the patients received vitamin D_3_ supplementation. Thus, the data presented here demonstrate that, even after the first treatment interval of 3 months, an equalization of the 25(OH)D_3_ and 24,25(OH)_2_D_3_ levels for the entire patient cohort over the study period was achieved, regardless of the season of the start of the study. Out of the 32 patients before the study entry, 17 (53%) patients had 25(OH)D_3_ concentrations below 50 nmol/L, which is the recommended threshold indicator of vitamin D deficiency in humans [3,8]. Fourteen of the sixteen patients who began the study in winter/spring had values below this threshold. After the first treatment interval of 3 months, only three (9.4%) patients of the total study group remained with values below that limit (Fisher’s exact test, *p* = 0.0003).

### 3.3. Vitamin D_3_ Metabolites in Relation to the ZA Treatment

The repeated measurements in ZA-treated and ZA-untreated patients resulted in different curves for the respective individual vitamin D_3_ metabolite during the study (Figure 3). The 25(OH)D_3_ and 24,25(OH)_2_D_3_ levels were significantly lower at study entry in comparison with the levels at the three subsequent study time points, but not significantly different between ZA-treated and ZA-untreated patients at all the time points (Figure 3a,b). Thus, repeated-measures ANOVA for these two-factor studies showed ZA treatment to be a non-significant source of variation (*p*-values of 0.219 and 0.240; Figure 3a,b) and the time interval to be a significant source of variation (*p*-values of 0.0001 and 0.0007; Figure 3a,b). In contrast, the 1,25(OH)_2_D_3_ levels did not statistically differ between the ZA-treated and ZA-untreated patients at any of the measuring points (Figure 3c). These data prove that ZA treatment did not alter the levels of the three metabolites during the study. It can be concluded that the differences in the levels of 25(OH)D_3_ and 24,25(OH)_2_D_3_ observed between the study’s start and the subsequent measuring points were due to the concomitant supplementation of vitamin D_3_ to all the study patients.

### 3.4. Vitamin D_3_ Metabolites in Relation to the Development of Bone Metastasis during the Study

The analysis of the concentrations of 25(OH)D_3_ and 24,25(OH)_2_D_3_ regarding metastasis showed that they corresponded with those observed under the aspect of the ZA treatment (Figure 4). Neither metabolite was associated with the development of bone metastases in patients during the study, as the factor “metastasis” was not a significant variable of the source of variation (Figure 4a,b). The time-dependent changes can also be attributed to the concomitant vitamin D_3_ supplementation. This is in contrast to the very striking 1,25(OH)_2_D_3_ profile of the patients who did or did not suffer from bone metastasis during the study (Figure 4c). The patients who developed bone metastasis already had higher 1,25(OH)_2_D_3_ values before the study’s start compared to those without bone metastasis. This pattern remained throughout the study period. This observation suggests that 1,25(OH)_2_D_3_ could be a possible factor associated with the metastatic process in PCa. Since our study was by no means designed to make prognostic statements, we have only compiled these indicative data in the supplement for interested readers (Appendix A).

### 3.5. Correlations between Vitamin D_3_ Metabolites

Strong correlations between the 25(OH)D_3_ and 24,25(OH)_2_D_3_ levels were observed at the four measuring points, with correlation coefficients between 0.696 and 0.883 (mean ± SD; 0.776 ± 0.087) and *p*-values of <0.0001 in all cases. In this respect, the so-called vitamin D metabolite ratio (VMR), calculated as the ratio of 24,25(OH)_2_D_3_/25(OH)D_3_ × 100, is of interest, as this ratio was suggested as an improved indicator of the vitamin D_3_ status [39]. The close correlation between the two metabolites explains that a similar pattern was observed as for the two individual metabolites (Figure 5).

In contrast to the strong association between 25(OH)D_3_ and 24;25(OH)_2_D_3_, the coefficients of the correlation between 25(OH)D_3_ and 1,25(OH)_2_D_3_ as well as between 24,25(OH)_2_D_3_ and 1,25(OH)_2_D_3_ at the four measuring points were all non-significant, with values between -0.122 and 0.145 (−0.004 ± 0.148; *p*-values of 0.266 to 0.915) and −0.175 and 0.051 (−0.044 ± 0.174; *p*-values of 0.337 to 0.916), respectively.

## 4. Discussion

Our study showed that the main vitamin D metabolites 25(OH)D_3_, 24,25(OH)_2_D_3_, and 1,25(OH)_2_D_3_ were not affected in high-risk PCa patients who received ZA as supportive-care treatment over about 4 years. The ZA-treated patients and controls without ZA, who had 25(OH)D_3_ concentrations below the deficiency threshold of 50 nmol/L [1,40] due to a study start in winter/spring, achieved stable levels above this limit after 3 months with a daily concomitant supplementation of 400–500 IU of vitamin D cholecalciferol. These data additionally indicate good patient compliance with the supplement administration in contrast to other reports [41]. Simultaneously, stable 24,25(OH)_2_D_3_ levels were also observed afterwards during the three subsequent time intervals. The occurrence of bone metastases also did not result in altered profiles for these two metabolites. The metabolite 1,25(OH)_2_D_3_ also did not show profile changes during the entire observation time, but it was completely unaffected by the cholecalciferol supplementation, in contrast to 25(OH)D_3_ and 24,25(OH)_2_D_3_. In addition, it was remarkable that patients with metastasis already had higher concentrations of 1,25(OH)_2_D_3_ in comparison to those patients without metastasis at the study’s beginning and during the entire study.

Thus, the results partly differed for 25(OH)D_3_ and 24,25(OH)_2_D_3_, on the one hand, and for 1,25(OH)_2_D_3_, on the other hand. Therefore, it is advisable to discuss the data for the metabolites separately.

It is currently generally accepted that the circulating 25(OH)D_3_ is the best indicator characterizing the vitamin D status [1]. However, a final consensus about the definition of vitamin D deficiency based on a cutoff level of 25(OH)D_3_ was not reached in the last International Conferences on Controversies in Vitamin D [1,2,3,8]. The Endocrine Task Force on Vitamin D defined a 25(OH)D_3_ level of 50 nmol/L as a deficiency cutoff [42]. This cutoff was also recommended by the Institute of Medicine, USA [40]. A higher threshold of 75 nmol/L was suggested by other expert groups [8]. This absence of consensus results mainly from the lack of traceability and harmonization/standardization of the various 25(OH)D_3_ assays that were applied in the different studies [3,43,44]. Our study revealed that, after treatment with vitamin D cholecalciferol, the high percentage of patients with deficient levels of 25(OH)D_3_ below 50 nmol/L at the start of the study could be reduced from 53 to 9.4%. In guidelines and comments, a daily supplement dosage of 10 to 50 µg (400–2000 IU) of vitamin D has been recommended to achieve at least this threshold of 50 nmol/L [8,40,45,46,47,48,49]. As the half-life of circulating 25(OH)D_3_ is estimated to be approximately 15 days [50], this daily supplementation results in a steady state of 25(OH)D_3_ after three to four months [48,51]. The increase in circulating 25(OH)D_3_ depends on the baseline level and the dose of the supplemented vitamin D. For an initial level of 25 nmol/L, an increase to more than 60 nmol/L was reported with a daily supplement of 400 IU for three months [52]. This corresponds with the observation in our study (Figure 2b). A similar pattern was visible for the metabolite 24,25(OH)_2_D_3_. The first and second hydroxylation steps converting 25(OH)D_3_ to 24,25(OH)_2_D_3_ and 1,25(OH)_2_D_3_, respectively, are likely to be inversely regulated by the same effectors (parathormone, 1,25(OH)_2_D_3_, and fibroblast growth factor 23/klotho) [53,54,55], but there remains a close relationship between the two metabolites 25(OH)D_3_ and 24,25(OH)_2_D_3_ in the bloodstream. This is reflected in the strong Pearson correlation coefficients of 0.776 during the entire study period. The VMR, calculated as the ratio between 24,25(OH)_2_D_3_ and 25(OH)D_3_, also confirms the increased levels of both metabolites after three months of the study compared with the baseline values at the study’s beginning (Figure 5). VMR has been proposed as a more sensitive indicator for monitoring vitamin D intake [56,57,58], but some recent studies failed to confirm this advantage over the assessment based on 25(OH)D_3_ only [39,59,60].

Several studies in osteoporosis patients treated with ZA or other bisphosphonates have shown that there is a close relationship between the observed increased bone-mineral density and the circulating 25(OH) vitamin D concentration [33,61,62,63,64]. To achieve this treatment effect, different threshold levels of 25(OH) vitamin D have been reported. This is certainly because the assays used in various studies did not show clear traceability [3,33]. However, despite these conflicting data, studies concerning the usefulness of ZA in PCa patients have generally been performed with a concomitant supply of vitamin D both in the trial and placebo arms [27,34]. Follow-up data for vitamin D metabolites, however, are lacking. The less-satisfactory evidence for vitamin D in combination with bisphosphonates has been summarized in a meta-analysis [65]. Out of 27 randomized studies [65], the authors of one of only three studies with ZA monotherapy without the administration of vitamin D advised the prophylactic administration of vitamin D and the monitoring of the vitamin D levels for these patients [23]. In this respect, our follow-up data for the vitamin D metabolites support this recommendation. The data show that, with daily medication with 100 to 125 µg of cholecalciferol, a long-term level of 25(OH)D_3_ of >50 nmol/L can be achieved. In recent PCa guidelines, ZA has been recommended as a bone-protective agent and for pain relief in castration-resistant PCa patients and those with bone metastases [29,30]. Thus, ZA continues to be an important component of PCa management, even though the primary expectations of preventing bone metastases were not met [30,34].

For 1,25(OH)_2_D_3_, the reference range (95% confidence interval) in the serum/plasma of healthy adults (between 20 and 70 years) has been determined to be 59 to 159 pmol/L [66]. The circulating 1,25(OH)_2_D_3_ accounts for only approximately 0.1% of 25(OH)D_3_. A comparable proportion of the two metabolites was detected in prostate tissue, and their concentrations were correlated with serum levels [67,68]. The baseline concentrations of 1,25(OH)_2_D_3_ in our PCa cohort were within this reference range, except for three patients with lower values. Moreover, the repeated measures in our study showed that the concentrations did not significantly change over the entire period. There were no increased 1,25(OH)_2_D_3_ values due to the supplementation of cholecalciferol, in contrast to 25(OH)D_3_ and 24,25(OH)_2_D_3_, particularly during the first treatment interval (Figure 2e,f) and for the subclassification with/without metastasis (Figure 4c). The circulating 1,25(OH)_2_D_3_ is strictly controlled by a multiregulatory feedback system consisting of parathormone, fibroblast growth factor, calcium, phosphate, and 1,25(OH)_2_D_3_ itself [69]. In consequence, normal circulating levels of 1,25(OH)_2_D_3_ are largely ensured by the adequate synthesis of 1,25(OH)_2_D_3_ from its precursor 25(OH)D_3_, even at moderately decreased concentrations of 25(OH)D_3_ [66,70]. This was also evident in our study and, likewise, explains the missing correlations between 1,25(OH)_2_D_3_ and 25(OH)D_3_ or 24,25(OH)_2_D_3_. Other studies with and without additional vitamin D intake also reported missing correlations or low coefficients for the correlation between 1,25(OH)_2_D_3_ and 25(OH)D_3_ [38,70,71,72,73,74,75]. The peculiarity of sufficiently functioning 1,25(OH)_2_D_3_ synthesis despite a limited 25(OH)D_3_ substrate supply as long as a severe vitamin deficiency is not present also makes it understandable that 1,25(OH)_2_D_3_ is not considered a valid marker for global vitamin D deficiency [70,76].

However, our finding of higher 1,25(OH)_2_D_3_ levels in patients with subsequent metastasis during the study compared with distinctly lower levels in patients without progression was apparently surprising. However, it should be pointed out that the higher values in the metastasized PCa group were always in the reference range of circulating 1,25(OH)_2_D_3_ [66]. Significantly, the elevated baseline values were confirmed during the study. This is in contrast to other PCa and cancer studies in which increased levels of circulating 1,25(OH)_2_D_3_ were associated with improved outcome data [20,67,77,78]. Numerous preclinical studies based on cell-culture experiments and animal studies showed that 1,25(OH)_2_D_3_ inhibits the proliferation, migration, and invasion of cancer cells; suppresses angiogenesis; activates the apoptosis and differentiation of cells; or synergistically potentiates the antitumor activity of chemotherapeutic agents [5,79,80,81,82,83,84,85,86]. Since 1,25(OH)_2_D_3_ is the actual active vitamin D metabolite, these experimental data are also used as arguments to confirm the hypothesis of an anticancer effect of vitamin D [5,87]. However, it is noticeable that the 1,25(OH)_2_D_3_ concentrations used in the experiments are often 100–1000-fold higher than those detected in the bloodstream and target tissues [67,88,89,90]. This obvious contradiction has largely been ignored in the literature to date [91]. Furthermore, other experiments with a transgenic prostate mouse model showed enhanced distant metastasis upon prolonged treatment with 1,25(OH)_2_D_3_ [92]. Increased metastasis in treatment experiments with 1,25(OH)_2_D_3_ was also observed in a model of mammary-gland cancer in mice depending on the age of the mice [93,94].

We interpret the higher 1,25(OH)_2_D_3_ in the subgroup of PCa patients with metastasis after prostatectomy as a possible reflection of the interrelated complex action of this vitamin metabolite. 1,25(OH)_2_D_3_ not only directly influences tumor development via the vitamin D receptor as mentioned above but also indirectly modulates this process through crosstalk with the tumor microenvironment, different immunological pathways, and the functional interplay between the vitamin D and androgen receptors [6,7,9,95]. It is also conceivable that the higher serum levels of 1,25(OH)_2_D_3_ in the case of the subsequently metastasized PCa subcohort led, through the C23 and C24 metabolic pathways for 1,25(OH)_2_D_3_, to higher levels of their intermediates in cells [96]. These intermediates, for which very little is yet known [69], could favor direct or indirect cancerogenesis-promoting effects. Obviously, studies on their possible molecular mechanisms require experiments with biologically relevant concentrations, as already critically discussed above. On the other hand, this association between higher 1,25(OH)_2_D_3_ levels and subsequent metastasis does not necessarily imply a causal relationship between the two observations. Due to the lack of corresponding follow-up data for the vitamin D metabolites in other studies, these particular results have likely not been captured to date. However, we think it is important to point out these findings so that they can be verified in other studies and provide potential prognostic decision support.

Some limitations of our study should be mentioned while interpreting the results. First, it was a retrospective study with a limited sample size of patients and without external validation. Second, only the three essential vitamin D metabolites could be measured due to the limited availability of the sample material. Third, all the patients in both the study and control arms received vitamin D and calcium. Despite these limitations, we consider the results of this study to provide interesting information for understanding open questions in the ongoing vitamin D debate in practice. The strength of our study is based on the use of sophisticated analytical methods with traceability and good analytical performance as well as the strict adherence to the requirements for valid vitamin D studies.

## 5. Conclusions

The two vitamin D metabolites 25(OH)D_3_ and 24,25(OH)_2_D_3_ were not affected by supportive ZA treatment or the development of metastasis over four years in our selected cohort of high-risk PCa patients after prostatectomy. Surprisingly, the low-abundance metabolite 1,25(OH)_2_D_3_ was already higher before the study’s start in patients who developed bone metastasis compared to those without bone metastasis. Before potential prognostic decision support can be provided, verification in other studies is necessary.

## Figures and Tables

**Figure 1 cancers-14-01560-f001:**
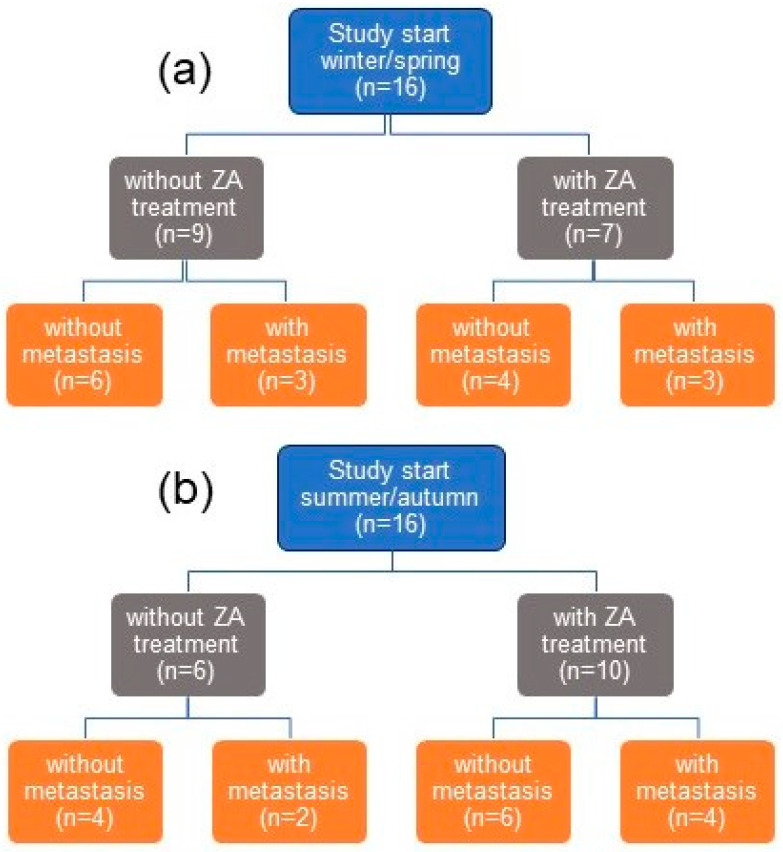
Overview of the study design and patient characteristics depending on the season of the study’s start in winter/spring (**a**) and summer/autumn (**b**). Abbreviation: ZA = zoledronic acid.

**Figure 2 cancers-14-01560-f002:**
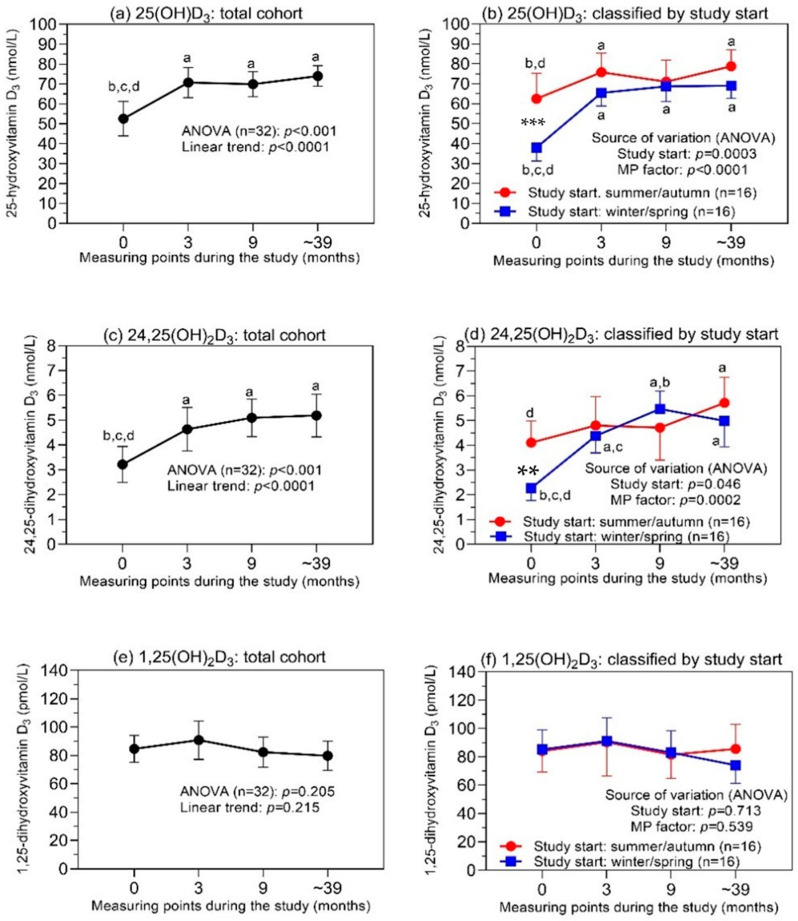
Levels of (**a**,**b**) 25(OH)D_3_, (**c**,**d**) 24,25(OH)_2_D_3_, and (**e**,**f**) 1,25(OH)_2_D_3_ at different time intervals of the study in the total cohort, and dependence on the season of the start of the study. Subfigures (**a**,**c**,**e**) present the results of the repeated-measures ANOVA for a single-factor study after treatment in the total cohort (*n* = 32). The corresponding subfigures (**b**,**d**,**f**) show the results of the two-factor study with repeated-measures ANOVA on the factor “study start” (winter/spring, *n* = 16, and summer/autumn, *n* = 16). Repeated measures were performed before the treatment (time point = 0) and 3 and 9 months after the treatment start. The last time point was 39 months (mean value) after the treatment start. Data at the time points are mean values with their 95% confidence intervals. At the error bars, the letters a, b, c, and d indicate statistically significant differences in the vitamin D_3_ levels between the different measuring points (at least *p* < 0.05; corrected values according to Holm–Sidak test): a, compared to “before study”; b, compared to 3 months; c, compared to 9 months; d, compared to ~39 months. Statistically significant differences between the metabolite levels of the two study subgroups at the respective time points are characterized by asterisks: **, *p* < 0.01; ***, *p* < 0.001. Abbreviations: ANOVA = analysis of variance; MP factor = related to the time intervals of the measuring points.

**Figure 3 cancers-14-01560-f003:**
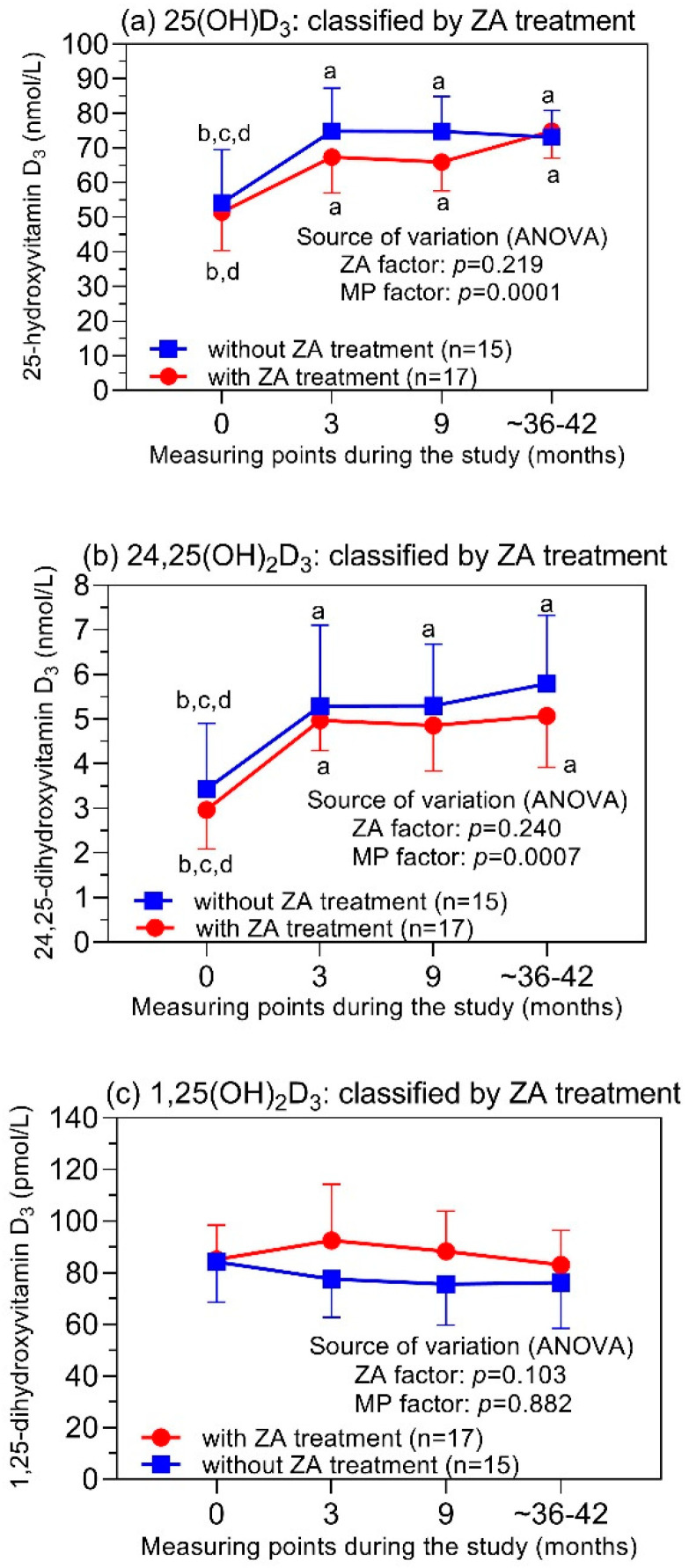
Levels of (**a**) 25(OH)D_3_, (**b**) 24,25(OH)_2_D_3_, and (**c**) 1,25(OH)_2_D_3_ before and during the ZA treatment at different time intervals of the study. Repeated measures were performed before the treatment (time point = 0) and 3 and 9 months after the treatment start. The last measuring points were 36 and 42 months (mean values) for patients without and with ZA treatment, respectively. Results of the repeated-measures two-factor ANOVA classified according to the factor ZA treatment (without ZA, *n* = 15; with ZA, *n* = 17) are shown as mean values with their 95% confidence intervals. At the error bars, the letters a, b, c, and d indicate statistically significant differences in the vitamin D_3_ levels between the different measuring points (at least *p* < 0.05; corrected values according to Holm–Sidak test): a, compared to “before study”; b, compared to 3 months; c, compared to 9 months; d, compared to ~36–42 months. No statistically significant differences for all three metabolite levels were found between the two ZA groups at the respective measuring points. Abbreviations: ANOVA = analysis of variance; ZA = zoledronic acid; MP factor = related to the time intervals of the measuring points.

**Figure 4 cancers-14-01560-f004:**
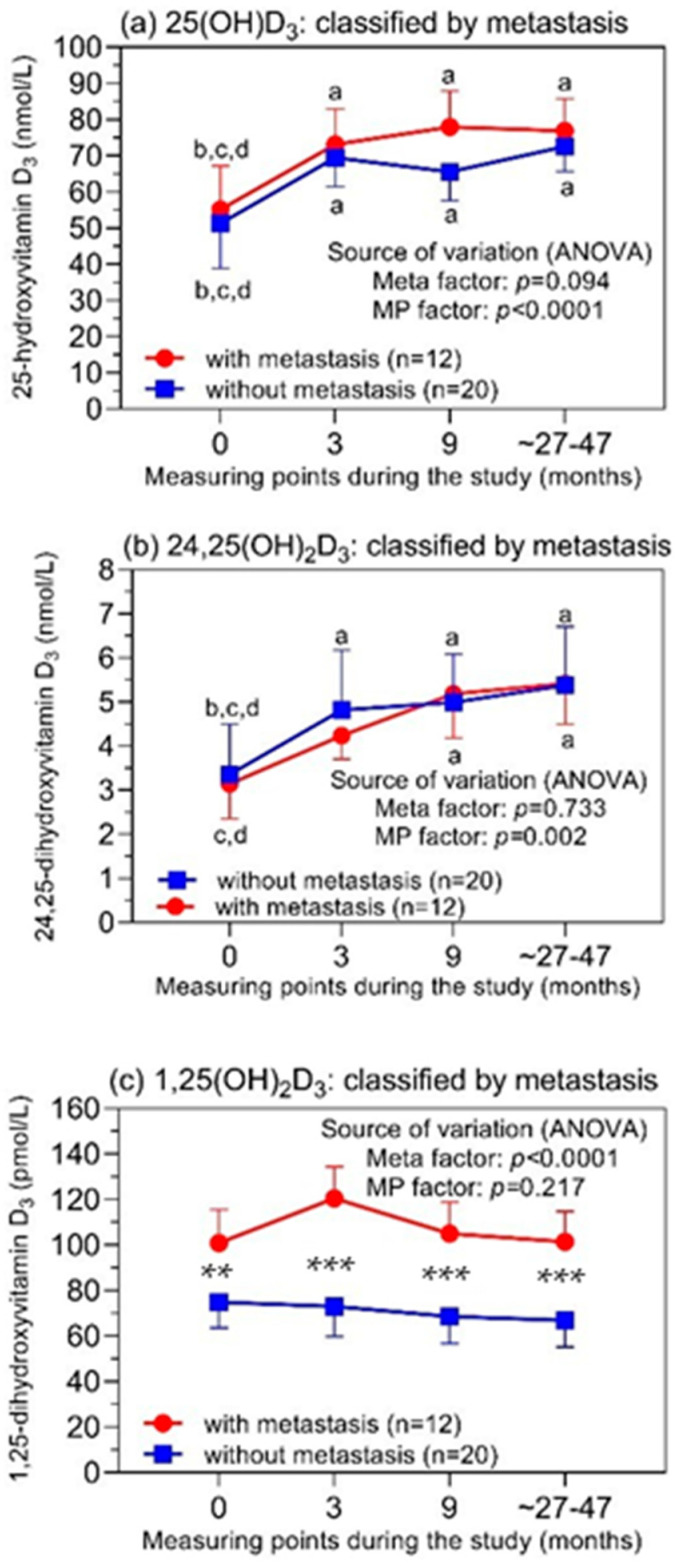
Levels of (**a**) 25(OH)D_3_, (**b**) 24,25(OH)_2_D_3_, and (**c**) 1,25(OH)_2_D_3_ in patients with and without developed bone metastases during the study. Repeated measures were performed before the treatment (time point = 0) and 3 and 9 months after the treatment start. The last measuring points were 27 and 42 months (mean values) for the patients with (*n* = 12) and without (*n* = 20) bone metastasis, respectively. Results of the repeated-measures two-factor ANOVA classified according to the factor metastasis are shown as mean values with their 95% confidence intervals. At the error bars, the letters a, b, c, and d indicate statistically significant differences in the vitamin D_3_ levels between the different measuring points (at least *p* < 0.05; corrected values according to Holm–Sidak test): a, compared to “before study”; b, compared to 3 months; c, compared to 9 months; d, compared to ~27–42 months. Statistically significant differences between the metabolite levels for the two study subgroups at the respective time points are characterized by asterisks: **, *p* < 0.01; *** *p* < 0.001. Abbreviations: ANOVA = analysis of variance; Meta factor = related to the developed bone metastasis; MP factor = related to the time intervals of the measuring points.

**Figure 5 cancers-14-01560-f005:**
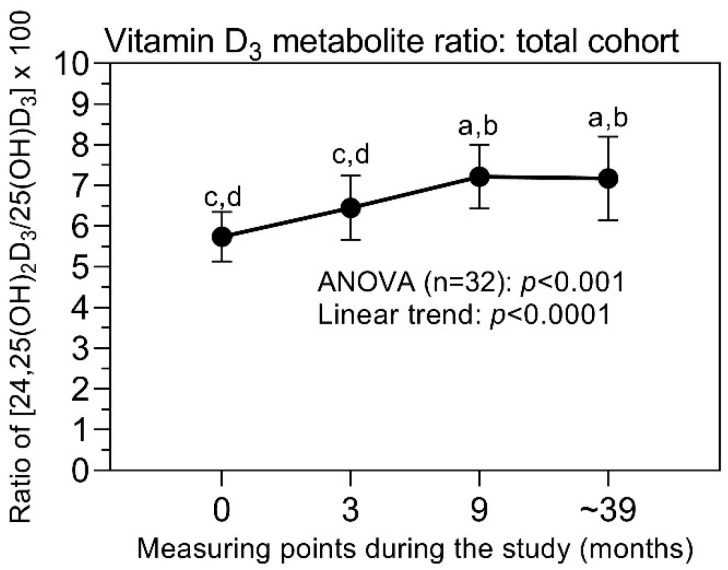
Change in the vitamin D_3_ metabolite ratio (VMR) in the total cohort during the study. The ratio [24,25(OH)_2_D_3_ to 25(OH)D_3_ × 100] defined as vitamin D_3_ metabolite ratio increased during the study. Data at the time points are mean values with their 95% confidence intervals. At the error bars, the letters a, b, c, and d indicate statistically significant differences between the levels at the different measuring points (at least *p* < 0.05; corrected values according to Holm–Sidak test): a, compared to “before study”; b, compared to 3 months; c, compared to 9 months; d, compared to ~39 months. Further explanations are provided in the legend of Figure 2.

## Data Availability

The data presented in this study are available upon reasonable request from the corresponding author.

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
