# Peer review of "Vitamin D Metabolites in Nonmetastatic High-Risk Prostate Cancer Patients with and without Zoledronic Acid Treatment after Prostatectomy"

_cancers, 2022, doi:10.3390/cancers14061560_

Round 1

Reviewer 1 Report

Comments to the Author:

This is a good study concerning the role of Vitamin D in prostate cancer patients. This is a controversial question, as the authors describe in the introduction section, with results considering the relationship between 25 OH vitamin D and prostate cancer, while other authors (among them the authors of this study) found no relationship between 25 OH Vitamin D and the aggressiveness of the tumour (Gleason score).

In this paper, the authors evaluated not only the concentration for 25 OH Vitamin D, but also their metabolites, 24,25 vitamin D and 1,25 OH vitamin D. Moreover, the analysis was performed by liquid chromatography-tandem mass spectrometry, offering a high quality of results. On the other hand, concerning the design of the study, the authors subdivided the patients into two groups in order to minimize seasonal differences on Vitamin D concentrations.

The paper is interesting and comprehensive and results well-summarized. The conclusion of this study is also remarkable, showing a relationship between 1,25 OH vitamin D and development of metastases, but not with 25 OH Vitamin D or 24,25 vitamin D.

This is an interesting paper for readers that, in my opinion, should be accepted for publication.

Reviewer 2 Report

I have read with great interest this manuscript, in which the authors provide new knowledge on the importance of vitamin D for the metastasis of prostate cancer. As there are many conflicting reports on this topic, this well-planned study can certainly initiate further research in this direction. This work shows how important it is not to focus on the basic metabolite of vitamin D in similar studies, but to broaden the panel. It seems especially important to test the level of calcitriol itself.

The methodology used by the authors to assess the levels of individual metabolites is also important. I have only one small note regarding the use of the word “subform” to describe vitamin D metabolites. Especially in the "simple summary", the use of the word is misleading. Better to use the word "metabolite". I did not find Fig. S1 in the materials provided.

Reviewer 3 Report

The present manuscript is well written and titled: “Vitamin D Metabolites in Nonmetastatic High-Risk Prostate Cancer Patients with and without Zoledronic Acid Treatment after Prostatectomy.” However, the manuscript lacks in vitro/ in vivo data to support the observations in the human cohort. Data needs to be refined in a presentable manner. The present manuscript would be benefited by addressing the points below.

Major comments:

  1. The scientific concept of the present manuscript is interesting. However, the scientific output from the study is minimal, and the sample size of the study is small. As the authors suggested, it needs to be conducted in the larger cohort to determine the prognostic value of vitamin-D in prostate cancer patients. 
  2. The authors need to provide some insight into the upregulation of vitamin-D in patients with bone metastasis in prostate cancer. The present manuscript lacks the molecular mechanism data to show how Vitamin-D affects prostate cancer metastasis.  
  3. The discussion section in the present manuscript is lengthy. Please consider condensing the discussion section in the manuscript. 

Round 2

Reviewer 3 Report

The authors have addressed all comments at their best. The manuscript is improvised after revision; the present manuscript is in acceptable form.